# Risperidone Treatment after Transient Ischemia Induces Hypothermia and Provides Neuroprotection in the Gerbil Hippocampus by Decreasing Oxidative Stress

**DOI:** 10.3390/ijms20184621

**Published:** 2019-09-18

**Authors:** Go Eun Yang, Hyun-Jin Tae, Tae-Kyeong Lee, Young Eun Park, Jeong Hwi Cho, Dae Won Kim, Joon Ha Park, Ji Hyeon Ahn, Sungwoo Ryoo, Young-Myeong Kim, Myoung Cheol Shin, Jun Hwi Cho, Choong-Hyun Lee, In Koo Hwang, Hui Jin, Moo-Ho Won, Jae-Chul Lee

**Affiliations:** 1Department of Radiology, Kangwon National University Hospital, Chuncheon, Gangwon 24289, Korea; yangke@kangwon.ac.kr; 2Bio-Safety Research Institute, College of Veterinary Medicine, Jeonbuk National University, Iksan, Jeollabuk 54596, Korea; anatotae@gmail.com (H.-J.T.); uribugi@naver.com (J.H.C.); 3Department of Neurobiology, School of Medicine, Kangwon National University, Chuncheon, Gangwon 24341, Korea; xorud312@naver.com (T.-K.L.); taeparo@naver.com (Y.E.P.); 4Department of Biochemistry and Molecular Biology, and Research Institute of Oral Sciences, College of Dentistry, Gangnung-Wonju National University, Gangneung, Gangwon 25457, Korea; kimdw@gwnu.ac.kr; 5Department of Anatomy, College of Korean Medicine, Dongguk University, Gyeongju, Gyeongbuk 38066, Korea; jh-park@kangwon.ac.kr; 6Department of Biomedical Science and Research Institute for Bioscience and Biotechnology, Hallym University, Chuncheon, Gangwon 24252, Korea; jh-ahn@hallym.ac.kr; 7Department of Biological Sciences, College of Natural Sciences, Kangwon National University, Chuncheon, Gangwon 24341, Korea; ryoosw08@kangwon.ac.kr; 8Department of Molecular and Cellular Biochemistry, School of Medicine, Kangwon National University, Chuncheon, Gangwon 24341, Korea; ymkim@kangwon.ac.kr; 9Department of Emergency Medicine, School of Medicine, Kangwon National University, Chuncheon, Gangwon 24341, Korea; dr10126@naver.com (M.C.S.); cjhemd@kangwon.ac.kr (J.H.C.); 10Department of Pharmacy, College of Pharmacy, Dankook University, Cheonan, Chungcheongnam 31116, Korea; anaphy@dankook.ac.kr; 11Department of Anatomy and Cell Biology, College of Veterinary Medicine, and Research Institute for Veterinary Science, Seoul National University, Seoul 08826, Korea; vetmed2@snu.ac.kr; 12Center for Nutraceutical and Pharmaceutical Materials, Myongji University, Yongin, Gyeonggi 17058, Korea; jimmykim83@mju.ac.kr

**Keywords:** ischemia/reperfusion, delayed neuronal death, antipsychotic drug, post-treatment, thermoregulation, 5-HT_2A_ antagonist

## Abstract

Compelling evidence from preclinical and clinical studies has shown that mild hypothermia is neuroprotective against ischemic stroke. We investigated the neuroprotective effect of post-risperidone (RIS) treatment against transient ischemic injury and its mechanisms in the gerbil brain. Transient ischemia (TI) was induced in the telencephalon by bilateral common carotid artery occlusion (BCCAO) for 5 min under normothermic condition (37 ± 0.2 °C). Treatment of RIS induced hypothermia until 12 h after TI in the TI-induced animals under uncontrolled body temperature (UBT) compared to that under controlled body temperature (CBT) (about 37 °C). Neuroprotective effect was statistically significant when we used 5 and 10 mg/kg doses (*p* < 0.05, respectively). In the RIS-treated TI group, many CA1 pyramidal neurons of the hippocampus survived under UBT compared to those under CBT. In this group under UBT, post-treatment with RIS to TI-induced animals markedly attenuated the activation of glial cells, an increase of oxidative stress markers [dihydroethidium, 8-hydroxy-2′ -deoxyguanosine (8-OHdG), and 4-Hydroxynonenal (4-HNE)], and a decrease of superoxide dismutase 2 (SOD2) in their CA1 pyramidal neurons. Furthermore, RIS-induced hypothermia was significantly interrupted by NBOH-2C-CN hydrochloride (a selective 5-HT_2A_ receptor agonist), but not bromocriptine mesylate (a D_2_ receptor agonist). Our findings indicate that RIS-induced hypothermia can effectively protect neuronal cell death from TI injury through attenuation of glial activation and maintenance of antioxidants, showing that 5-HT_2A_ receptor is involved in RIS-induced hypothermia. Therefore, RIS could be introduced to reduce body temperature rapidly and might be applied to patients for hypothermic therapy following ischemic stroke.

## 1. Introduction

Neuroprotective compounds provide protection in models of ischemic stroke; however, none have shown efficacy in clinical trials [1]. Body temperature influences outcome of ischemic stroke [2,3], and hypothermia is one of the most promising neuroprotective therapies, as assessed by the Stroke Therapy Academic Industry Roundtable Criteria [4]. Although the precise mechanism of hypothermic neuroprotection in ischemic stroke is not known, hypothermia may act upon multiple pathways to ultimately prevent neuronal death [5]. It has been attempted to find drugs, including atypical antipsychotic agents with hypothermic effects [6,7], to treat ischemic stroke. The drugs reduce temperature instantly when a patient is picked up by emergency personnel, which are already approved for patient treatment and used anywhere, independent of sophisticated cooling equipment, and may prolong therapeutic time window for other treatments. Thus, atypical antipsychotic agent-induced hypothermia may help to overcome brain injury following ischemic stroke. For several decades, risperidone (RIS) has been widely used for treatment of schizophrenia as a selective monoaminergic antagonist with high-affinity for serotonin type 2 (5-HT_2A_) receptor and dopamine type 2 (D_2_) receptor in the limbic system [8,9]. In addition, RIS-induced hypothermia has been reported [10,11]. It remains possible that RIS contains neuroprotection against ischemic stroke via hypothermia. Thus, this study determined whether neuroprotective effect of RIS is hypothermia as a post-conditioning stimulus for inducing ischemic tolerance and elucidated RIS-induced hypothermic mechanisms. For this purpose, we intended (1) to compare effects of RIS on transient ischemic damage between ischemic animals without regulating body temperature and with maintaining body temperature; (2) to study effects of RIS against ischemia-induced oxidative stress; and (3) to investigate roles of central 5-HT_2A_ and D_2_ receptors in body temperature regulation in RIS-treated animals.

## 2. Results

### 2.1. Effects of RIS Against TI Injury Under Uncontrolled Body Temperature (UBT)

#### 2.1.1. Body Temperature under UBT Condition after TI

Under UBT condition, an abrupt elevation of body temperature was seen in the TI + vehicle group after TI, and the maximum temperature (39 ± 0.5 °C) was at 1 h after TI; thereafter, body temperature was gradually decreased (Figure 1A). Unlike the TI + vehicle group, in the TI + RIS groups under UBT condition, body temperature was decreased, as shown in Figure 1A. RIS produced a dose-dependent decrease in body temperature, and the effect was statistically significant in 5 and 10 mg/kg doses (*p* < 0.05, respectively) in comparison to the TI + vehicle group (Figure 1A). In particular, the time course shows a rapid onset in the highest dose, with a maximal effect between 1 and 2 h and duration of 6 h. A similar pattern, although less pronounced, was seen with the two lower doses.

#### 2.1.2. Neuronal Nuclear Antigen (NeuN) Positive (^+^) and Fluoro-Jade B (F-J B)^+^ Neurons

The protection afforded by RIS against TI-induced neuronal death in the CA1 was assessed under UBT condition using NeuN immunohistochemistry and F-J B histofluorescence staining (Figure 1B–D). In the sham + vehicle group, pyramidal cells or neurons in the CA1, which are called CA1 pyramidal neurons, were well stained with NeuN; however, no F-J B^+^ CA1 pyramidal cells were found. At 5 days after TI, NeuN^+^ CA1 pyramidal neurons were markedly decreased, and many F-J B^+^ CA1 pyramidal cells were detected. In the sham + RIS group, CA1 pyramidal neurons were also well stained with NeuN, and F-J B^+^ cells were not observed. In the TI + RIS group, RIS produced a dose-dependent increase in the number of NeuN^+^ CA1 pyramidal neurons, and a dose-dependent decrease in the number of F-J B^+^ CA1 pyramidal cells 5 days after TI: The results were statistically significant for 5 and 10 mg/kg doses (*p* < 0.05, respectively) in comparison with the TI + vehicle group. Based on these findings, we found better results in the TI + RIS (10 mg/kg) group than those in the TI + RIS (5 mg/kg) group, and examined the underlying hypothermic mechanism related to the neuroprotective effect of RIS against TI damage in the TI + RIS (10 mg/kg) group.

#### 2.1.3. Glial Fibrillary Acidic Protein (GFAP)^+^ and Ionized Calcium-Binding Adapter Molecule 1 (Iba-1)^+^ Cells

Changes in GFAP^+^ and Iba-1^+^ cells in the CA1 were examined at 1 day and 5 days after TI (Figure 2A–D). GFAP^+^ and Iba-1^+^ cells, as a resting for, were distributed throughout all layers of the CA1 of the sham + vehicle group. In the TI + vehicle group, immunoreactivity of the GFAP and Iba-1 was slightly increased 1 day after TI compared to that in the sham + vehicle group. Five days after TI, the immunoreactivity was significantly increased, and most of the GFAP^+^ and Iba-1^+^ cells were hypertrophied in the CA1. In the sham + RIS group, the immunoreactivity and morphology of GFAP^+^ and Iba-1^+^ cells were similar to that in the sham + vehicle group. In the TI + RIS group at 1 day after TI, the immunoreactivity and morphology of GFAP^+^ and Iba-1^+^ cells were not significantly changed compared to the sham + vehicle group. However, 5 days after TI, these immunoreactivity in the TI + RIS group was significantly lower than that in the TI + vehicle group.

### 2.2. Abolishment of RIS-Mediated Neuroprotection under Controlled Body Temperature (CBT)

#### 2.2.1. Body Temperature under CBT Condition after TI

After the onset of TI, body temperature was maintained within normothermic range (37.0–37.3 °C) with a feedback-regulated heating pad for 12 h (Figure 3A). An abrupt elevation of body temperature was seen in the TI + vehicle group at 1 h after TI under CBT condition: The temperature was intensely increased (approximately 39.1 ± 0.5 °C). Thereafter, body temperature was gradually decreased. However, under CBT condition, the degree of hypothermia induced by 5 and 10 mg/kg RIS injection was higher, respectively, than that under UBT condition, showing that body temperature was 36.8 and 36.1 °C, respectively, at 1 h after the first RIS treatment, and 36.6 and 36.1 °C, respectively, after the second injection.

#### 2.2.2. NeuN^+^ Neurons, F-J B^+^ Cells, GFAP^+^ Astrocytes, and Iba-1^+^ Microglia

Effects of CBT on RIS-mediated neuroprotection are shown in Figure 3B,C. In the sham + RIS group under CBT, the distribution of NeuN^+^ CA1 pyramidal neurons was similar to that in the sham + vehicle group. In the TI + RIS group under CBT, NeuN^+^ CA1 pyramidal neurons were significantly decreased 5 days after TI. In addition, many F-J B^+^ CA1 pyramidal cells were shown 5 days after TI under CBT condition. Changes of GFAP^+^ and Iba-1^+^ cells under CBT condition were examined in the CA1 5 days after TI (Figure 3B,D). In the sham + RIS group under CBT, the immunoreactivity and morphology of GFAP^+^ and Iba-1^+^ cells were not significantly different from those of the sham + vehicle group. In the TI + RIS group under CBT, GFAP and Iba-1 immunoreactivity 5 days after TI was significantly increased in the CA1, like the TI + vehicle group under UBT condition.

### 2.3. Effects of RIS Against TI-Induced Oxidative Stress under UBT and CBT Conditions

#### 2.3.1. Superoxide Anion Production

To analyze the potential role of superoxide anion in RIS-mediated neuroprotection against TI injury, the production of superoxide anion in the CA1 pyramidal neurons was determined using DHE reactivity after TI (Figure 4A,B). DHE reactivity was similar in all sham groups. Under UBT condition, superoxide anion was intensely induced in CA1 pyramidal neurons 1 day after TI; however, the reactivity was significantly decreased in the TI + RIS group. At this time, under CBT condition, superoxide anion production in the TI + RIS group was similar to that in the TI + vehicle group under UBT condition. Five days after TI, DHE reactivity in CA1 pyramidal neurons was hardly shown in all groups, because CA1 pyramidal neurons were killed in the TI + vehicle and TI + RIS groups under CBT condition, and CA1 pyramidal cells in the TI + RIS under UBT condition survived and did not product superoxide anion.

#### 2.3.2. 8-Hydroxy-2′-deoxyguanosine (8-OHdG) and 4-hydroxy-2-nonenal (4-HNE) Immunoreactivity

Changes in 8-OHdG (oxidative DNA damage) and 4-HNE (lipid peroxidation) in the CA1 were examined at 1 day and 5 days after TI (Figure 5A–D). Very weak 8-OHdG and 4-HNE immunoreactivity was detected in CA1 pyramidal neurons in all sham groups. At 1 day after TI, in the TI + vehicle group under UBT, immunoreactivity of the 8-OHdG and 4-HNE in CA1 pyramidal neurons was dramatically increased. At this point in time, these immunoreactivity in the TI + RIS group under UBT was slightly increased; however, in the TI + RIS group under CBT, immunoreactivity of the 8-OHdG and 4-HNE was similar to that in the TI + vehicle group under UBT condition. Five days after TI, immunoreactivity of the 8-OHdG and 4-HNE in CA1 pyramidal neurons of the TI + vehicle groups disappeared due to death of CA1 pyramidal neurons. At this point in time, in the TI + RIS group under UBT, the immunoreactivity in CA1 pyramidal neurons was similar to that at 1 day after TI, and 8-OHdG and 4-HNE immunoreactivity of the TI + RIS under CBT was hardly shown because CA1 pyramidal neurons were killed under CBT condition. In the TI + vehicle group under UBT, 8-OHdG immunoreactivity was significantly increased in CA1 pyramidal neurons 1 day after TI, and, at 5 days after TI, 8-OHdG immunoreactivity was hardly shown due to loss of CA1 pyramidal neurons.

#### 2.3.3. Superoxide Dismutase 2 (SOD2) Level and Immunoreactivity

SOD2 protein levels in the CA1 were differently changed according to body temperature condition (Figure 6A). SOD2 protein levels in all sham groups were not significantly different between the groups. Under UBT condition, SOD2 level of the TI + vehicle group was dramatically decreased 1 day (41% of the sham group) after TI, and more decreased 5 days (22% of the sham group) after TI. In the TI + RIS group, SOD2 level was not significantly changed at any time after TI. However, under CBT condition, the change pattern of SOD2 level was similar to that in the TI + vehicle group (54% and 12% of the sham group). SOD2 immunoreactivity in CA1 pyramidal neurons was differently altered according to body temperature condition (Figure 6B). In all sham groups, SOD2 immunoreactivity in CA1 pyramidal neurons was similar. Under UBT, SOD2 immunoreactivity in the TI + vehicle group was significantly reduced 1 day after TI. At 5 days after TI, SOD2 immunoreactivity was more decreased; in particular, many non-pyramidal cells showed strong SOD2 immunoreactivity. In the TI + RIS group, SOD2 immunoreactivity in CA1 pyramidal neurons was maintained at all times after TI. Under CBT condition, the change pattern of SOD2 immunoreactivity in the CA1 of the TI + RIS group was like that in the TI + vehicle group.

### 2.4. Effects of 5-HT_2A_- and D_2_- Receptors on RIS-Induced Hypothermia

As shown in Figure 7, basal body temperature in all sham groups was homogenous (36.9 to 37.2 °C). Treatment with RIS (10 mg/kg) produced significant reduction (31.6 ± 0.91 °C) of body temperature at 60 min after the administration and slowly increased thereafter, and the hypothermic effect disappeared completely 240 min after the administration. NBOH-2C-CN hydrochloride (5 mg/kg) treatment did not significantly change body temperature, but bromocriptine mesylate (5 mg/kg) treatment showed significant decrease at 80 min after the administration (32.9 ± 1.64 °C), and the hypothermic effect disappeared 240 min after the administration. On the other hand, (±)-DOI hydrochloride (5 mg/kg) treatment slightly increased body temperature at 60 min after the administration (38.6 ± 0.93 °C).

To find out roles of endogenous 5-HT_2A_- and D_2_- receptors in RIS-induced hypothermic effect, animals were pre-treated with NBOH-2C-CN hydrochloride, (±)-DOI hydrochloride, and bromocriptine mesylate before RIS treatment. Significant inhibition of RIS-induced hypothermia was observed in the animals treated with NBOH-2C-CN hydrochloride and (±)-DOI hydrochloride in comparison to the animals treated with RIS alone. Unlike the 5-HT_2A_ receptor agonist, the hypothermic effect produced by RIS was significantly aggravated by bromocriptine mesylate. This finding might conclude that hypothermia induced by RIS could be a consequence of 5-HT_2A_ receptor antagonism.

## 3. Discussion

TI selectively induces delayed neuronal death (DND) in CA1 pyramidal neurons [12]. Recently, we reported that treatment with RIS dose-dependently protected against TI-induced DND of CA1 pyramidal neurons in the gerbil hippocampus [13]. In this study, we compared neuroprotection of RIS against TI under UBT and CBT and found that 10 mg/kg RIS post-treatment significantly attenuated the DND of CA1 pyramidal neurons under UBT; however, RIS did not protect the neurons under CBT after TI. This finding suggests that a critical interaction between body temperature and RIS appears to permit or prevent neuroprotection. 

Hypothermia is clearly established as an effective protectant against neuronal damage/death after TI [5]. In humans, hypothermia also serves as a neuroprotective procedure against cerebral hypoxia during neurosurgery and cardiovascular surgery [14,15]. Optimal protection is typically gained when hypothermia is induced as early as possible after stroke onset, with a mild-to-moderate hypothermia (32 to 35 °C) that lasts at least 1 to 2 h [16]. From laboratory studies, it is clear that hypothermia more consistently protects hippocampal neurons when applied soon after or even before the onset of TI [17,18]. Therefore, therapeutic hypothermia should be initiated as soon as possible to achieve its optimal beneficial effect. The American Heart Association cardiopulmonary resuscitation guidelines recommends that the duration of hypothermia should be at least 12 h [19]. Common methods for inducing therapeutic hypothermia are surface cooling and pharmacological cooling; each offers unique advantages and disadvantages [20]. In most clinical studies, therapeutic hypothermia is induced by body surface cooling. Unfortunately, the forced cooling method is slow and cumbersome, typically requiring several hours to reach target core temperature, and must be closely monitored to ensure the achievement of target temperature [21]. Furthermore, with surface cooling, precise control of core temperature is difficult. Recent methods using pharmacological agents have been suggested as a more efficient and faster way to reduce core temperature than surface cooling [22]. RIS has been reported to display hypothermia [6,10]. Therefore, this property makes RIS to recommend an ideal candidate for pharmacological agent as a more efficient and safer way for reducing body core temperature against ischemic insults. We, in this study, demonstrated that RIS treatment induced hypothermia within 30 min after post-TI and was maintained for 6 h. Although we originally planned to use an automated hypothermia retaining system to maintain hypothermia for more than 24 h, it was limited to 12 h because it was difficult to sustain due to practical problems. However, based on our study, we suggest that RIS treatment after post-TI might be easier to achieve hypothermia in the whole body and must be more practical in the clinic.

It is well known that glial cells, including astrocytes and microglia, are activated by ischemic stroke, and the activated glia has been suggested to contribute to delayed neuronal death, presumably via releasing neurotoxic substances, including reactive oxygen intermediates and pro-inflammatory cytokines [23,24,25]. To visualize glial cells in the hippocampal CA1 region induced by TI, we chose to use immunohistochemical staining for anti-Iba-1 (microglia) and anti-GFAP (astrocytes). We found that the activation of GFAP^+^ and Iba-1^+^ cells after post-treatment with RIS was decreased significantly at 1 and 5 days after ischemia-reperfusion under UBT condition compared CBT condition. While resident glial cells exist in a ramified state, after brain injury they migrate toward the lesion, their cell body becomes ameboid-shaped, the processes shorten, and become virtually indistinguishable from macrophages. Consistent with this, in vitro, hypothermia inhibits microglia proliferation, and attenuated microglia neurotoxicity, during and critically, after exposure to hypoxia and lipopolysaccharide [26,27,28]. Post-ischemic hypothermia also suppressed activated microglia after cerebral ischemia in fetal sheep [29]. A recent study reported that RIS modulated morphology and functions of glial cells in C6 astroglial cells model [30]. Also, it was reported that RIS significantly inhibited the production of nitric oxide and pro-inflammatory cytokines by interferon-γ-activated microglia [31]. Therefore, in the present study, RIS-induced mild hypothermia also reduced the activation and proliferation or glial cells after ischemia-reperfusion, and thereby possibly mitigated further pathological events, leading to cell death.

Oxidative stress leads to cell death because the stress causes widespread damage of cellular components and ultimately promotes cellular death after TI injury [32,33]. Ischemic insults produce ROSs, and the interaction between ROSs and DNA produces DNA strand break and base modification, which are frequently assessed by measurement of nucleoside 8-OhdG level. Here, we compared oxidative stress between the TI + vehicle and TI + RIS group. DHE fluorescence, 8-OhdG, and 4-HNE immunoreactivity were significantly increased in CA1 pyramidal neurons of the TI + vehicle group and apparently decreased by RIS treatment under UBT after TI; the effect was abolished under CBT after TI. Antipsychotic drugs including RIS have a capacity to elevate antioxidants and could be used for therapy of oxidative stress that induces neuronal damage [34,35]. Antioxidants have protective effects against cerebral ischemia, suggesting that antioxidants are involved in the control of cellular damage after ischemic insults [36,37]. In this context, Altinkilic et al. [34] have demonstrated that RIS has a capacity to elevate antioxidants and is used for therapy of oxidative stress. In addition, Yan et al. [13] have reporetd that RIS treatment well maintains protein level and immunoreactivity of SOD2 in the CA1 induced by TI. In this study, level and immunoreactivity of SOD2 in the TI + RIS group was maintained under UBT, but significantly decreased under CBT. Therefore, we represent that RIS-induced hypothermia protects neurons from TI damage via maintaining endogenous antioxidants.

Alternatively, regulated hypothermia by downregulation of endogenous thermoregulation may be a better method to achieve therapeutic efficacy without severe adverse events. RIS has strong affinity for 5-HT_2A_ and D_2_ receptor with hypothermic regulatory action in the brain, and it is defined as a drop in core body temperature below 35 °C [6,11]. In this regard, we hypothesized that 5-HT_2A_ and receptor D_2_ antagonism play roles in RIS-mediated hypothermia, and found that RIS-induced hypothermia was blocked by (±)-DOI hydrochloride or NBOH-2C-CN hydrochloride treatment, suggesting that (±)-DOI hydrochloride may be involved in RIS-mediated thermoregulation. However, it was difficult to use (±)-DOI hydrochloride because it produced hyperthermia in the sham + vehicle animals. (±)-DOI hydrochloride induces hyperthermic response via a 5-HT_2A_-mediated mechanism in rats [38,39]. Instead, we found that NBOH-2C-CN hydrochloride did not cause hyperthermia in the vehicle animals, and RIS-induced hypothermia was partially blocked by NBOH-2C-CN hydrochloride. Based on our findings, we suggest that blockage of 5-HT_2A_ receptor is essential for RIS-induced hypothermia. Bromocriptine mesylate, a potent agonist at D_2_ receptor, could be used to control central hyperthermia [40]. Furthermore, RIS has stronger affinity for 5-HT_2A_ receptor than for D_2_ receptor [6,41]. Bromocriptine mesylate induces hypothermia and reduces neuronal damage after global ischemia in rats [42]. We, in this study, found that bromocriptine mesylate significantly lowered body temperature in the TI + vehicle group and further lowered RIS-mediated hypothermia. From the reports and our data, we suggest that D_2_ receptor might be of minor significance since RIS-induced hypothermia did not recur with a subsequent D_2_ agonist treatment. Therefore, the hypothermic effect of RIS leads us to conclude that RIS-induced hypothermia might be mediated by 5-HT_2A_ receptor antagonism rather than by D_2_ receptor antagonism.

In conclusion, RIS-induced hypothermia protected neurons from TI, and the neuroprotection might be associated with attenuation of glial activation and maintenance of antioxidants following hypothermia, showing that the hypothermia was abolished by NBOH-2C-CN hydrochloride treatment. These suggest that 5-HT_2A_ receptor is involved in RIS-mediated reduction of body temperature, and RIS-induced hypothermia as a post-conditioning stimulus might be given to stroke patients immediately at pick-up for hospitalization.

## 4. Materials and Methods

### 4.1. Animals

Male Mongolian gerbils (total number = 480) at 6 months of age (body weight 65–75 g) were obtained from the Experimental Animal Center, Kangwon National University, Chuncheon, Republic of Korea. As we described previously [43], the experimental protocol of this study was approved (no. KW-160802-1) by the Institutional Animal Care and Use Committee (IACUC) at Kangwon University. The experimental protocol adhered to guidelines from the current international laws and policies [43].

### 4.2. Experimental Groups, Induction of TI, and RIS Treatment

Experiments related with TI were two (experiment I and II). Animals in experiment I were subjected to 5 min TI, and body temperature of the animals was uncontrolled for 12 h after TI. Meanwhile, in experiment II, animals were subjected to 5 min TI, and body temperature of the animals was controlled (37 ± 0.2 °C) for 12 h after TI. Animals (total number = 432) in each experiment were divided into four groups (*n* = 7 at each point in time in each group) as follows: (1) Sham + vehicle group, which was given sham surgery for 5 min TI and intraperitoneally injected with vehicle; (2) TI + vehicle group, which was given 5 min TI and intraperitoneally injected with vehicle; (3) sham + RIS group, which was subjected to sham TI and intraperitoneally injected with RIS; and (4) TI + RIS group, which was subjected to 5 min and treated with RIS.

TI was developed as previously described [44]. In brief, the animals were anesthetized with a mixture of 2.5% isoflurane in 32% oxygen and 68% nitrous oxide. Both common carotid arteries were occluded using non-traumatic aneurysm clips (Yasargil FE 723K, Aesculap, Tuttlingen, Germany). The complete stop of blood circulation was examined in central arteries in both retinae with an ophthalmoscope (HEINE K180^®^, Heine Optotechnik, Herrsching, Germany). The body temperature (37 ± 0.2 °C) was controlled using a thermometric blanket before and during the surgery. After 5 min of occlusion, the aneurysm clips were removed.

Vehicle or RIS (1, 5, and 10 mg/kg; Sigma-Aldrich, St. Louis, MO, USA) was intraperitoneally administered two times (immediately and at 6 h) after TI operation. Doses of RIS were selected based on a previous study [45]. RIS was dissolved in 0.3% Tween 80 in saline.

For recording temperature change in the body, rectal temperature was measured every 1 h after TI, over a 12 h period, in a room with ambient temperature of 22  ±  1 °C. The animals in all the groups were given recovery times of 5 days after TI, because pyramidal neurons in the hippocampal CA1 area do not die until 3 days and begin to die 4 days after TI [44,46,47].

### 4.3. Histological Tissue Preparation

As previously described [44], animals (*n* = 6 at each point in time in each group) were anesthetized with pentobarbital sodium (30 mg/kg; JW Pharmaceutical, Seoul, Korea) at the designated times and perfused transcardially with 0.1 M phosphate-buffered saline (PBS, pH 7.4) followed by 4% paraformaldehyde in 0.1 M phosphate-buffer (PB, pH 7.4). Brain tissues containing hippocampi were serially sectioned into 30 μm coronal sections in a cryostat (CM1900 UV, Leica, Wetzlar, Germany).

### 4.4. F-J B Histofluorescence Staining

For neuronal degeneration, F-J B (a high affinity fluorescent marker for neurodegeneration) staining was performed according to our published procedure [44]. Briefly, the sections were immersed in a solution containing 1% sodium hydroxide, transferred to a solution of 0.06% potassium permanganate, and then a solution of 0.0004% Fluoro-Jade B (Histochem, Jefferson, AR, USA). After washing, the incubated sections were placed on a slide warmer (approximately 50 °C) to be reacted. Finally, we examined the sections using an epifluorescent microscope (Carl Zeiss, Göttingen, Germany) with blue (450–490 nm) excitation light and a barrier filter.

### 4.5. Superoxide Anion Production Detection

For the analysis of oxidative stress, an oxidative fluorescent dye, dihydroethidium (DHE; Sigma-Aldrich), was used to evaluate in situ production of superoxide anion. Histological detection of superoxide anion was performed as described previously [48] at sham, 1 day and 5 days after TI (*n* = 7 at each time in each group). In brief, the sections were incubated with DHE (10 μmol/L) in PBS for 30 min at 37 °C in a humidified chamber that was shielded from light. DHE was oxidized on reaction with superoxide to ethidium, which could bind to DNA in the nucleus and fluoresced red. For the detection of ethidium, the reacted sections were examined with an epifluorescent microscope (Carl Zeiss) with an excitation wavelength of 520–540 nm. The fluorescence intensity of the sections was analyzed in the stratum pyramidale at the center of the CA1 field.

### 4.6. Immunohistochemistry

Immunohistochemistry was carried out according to our published procedure [43]. In brief, the sections were incubated with primary mouse anti-NeuN (a neuron-specific soluble nuclear antigen, diluted 1:1000, Chemicon International, Temecula, CA, USA) for neurons, mouse anti-GFAP (diluted 1:1000, Chemicon International) for astrocytes, rabbit anti-Iba-1 (1:800, Wako, Osaka, Japan) for microglia, mouse anti-4-HNE (diluted 1:1000; Alexis Biochemicals, San Diego, CA, USA) for lipid peroxidation [49], goat anti-(8-OhdG) diluted 1:500; EMD Millipore Corporation, Billerica, MA, USA) for DNA damage [50,51], and sheep anti-SOD2 (1:1000; Calbiochem, San Diego, CA, USA) for oxidative stress. The incubated sections were reacted with corresponding secondary antibodies (Vector Laboratories Inc., Burlingame, CA, USA), and developed using Vectastain ABC (Vector Laboratories Inc.). Finally, the immunoreacted sections were visualized with 3,3′-diaminobenzidine.

### 4.7. Western Blotting for SOD2

To obtain exact data for alterations in levels of SOD2 protein in the CA1 field after TI, animals (*n* = 7 at each point in time) were killed at designated times (sham, 1 day and 5 days) after TI, and tissues containing the CA1 field were used for Western blot analysis as previously described [44].

### 4.8. Effects of 5-HT_2A_ and D_2_ Agonists against RIS-Induced Hypothermia

To examine the contribution of 5-HT_2A_ or D_2_ antagonism to RIS-induced hypothermia, additional gerbils (*n* = 6/group) were intraperitoneally administered with (±)-2,5-dimethoxy-4-iodoamphetamine hydrochloride ((±)-DOI, 5-HT_2A/2C_ agonist, 5.0 mg/kg, Sigma-Aldrich), 4-[2-[(2-hydroxyphenyl)methylamino]ethyl]-2,5-dimethoxybenzonitrile (NBOH-2C-CN) hydrochloride (a selective 5-HT_2A_ agonist, 5.0 mg/kg, Tocris Bioscience, Ellisville, MO, USA), or bromocriptine mesylate (a D_2_ agonist, 5.0 mg/kg, Sigma-Aldrich) with vehicle or RIS (10.0 mg/kg), simultaneously. Rectal temperature was measured every 20 min after drug administration, over a 240 min period.

### 4.9. Data Analysis

Data analyses were performed by two or three investigators, who were blind to the experimental conditions. To analyze cell numbers and immunoreactivities, we selected six sections/animal with 120 µm interval according to anatomical landmarks corresponding to AP (anteroposterior) −1.65 ~ −3.40 mm of the gerbil brain atlas [52]. Firstly, we analyzed cell counts by averaging total numbers of NeuN-immunoreactive and F-J B-positive cells. Briefly, as previously described [44], images of the cells were captured in a 200 μm × 200 μm square at the center of the stratum pyramidale of the CA1 field using an AxioM1 light microscope (Carl Zeiss, Göttingen, Germany) equipped with a digital camera (Axiocam, Carl Zeiss) connected to a PC monitor. A ratio of the count was calibrated as % of the sham + vehicle group (NeuN-immunoreactive cells) or TI + vehicle group (F-J B-positive cells) using an image analyzing system (software: Optimas 6.5, CyberMetrics, Scottsdale, AZ, USA). Secondly, quantitative analyses of GFAP, Iba-1, 8-OHdG, 4-HNE, and SOD2 immunoreactivities and DHE fluorescence intensity in the CA1 area were done as previously described [44]. In short, staining intensities of GFAP, Iba-1, 8-OHdG, 4-HNE-, and SOD2 immunoreactive structures and superoxide anion radical were evaluated on the basis of an optical density (OD), which was obtained after the transformation of the mean gray level using the formula: OD = log (256/mean gray level). Background density in the images was subtracted, and brightness and contrast were calibrated as % (relative optical density, ROD) using Adobe Photoshop version 8.0 and analyzed using NIH Image J software. A ratio of the ROD was calibrated as %, with sham + vehicle group designated as 100%.

In addition, according to our previous method [44], we scanned results of Western blot bands and carried out densitometric analyses for the quantification of the bands using Scion Image software (Scion Corp., Frederick, MD, USA). Expression rates of the target proteins were normalized through corresponding expression rates of β-actin.

### 4.10. Statistical Analysis

The estimation of sample size was dependent on the standard deviation as in a published study by Ozkan et al. [53]. Sample size was at least seven gerbils per group with an alpha error of 0.05 and a power of > 80%, calculated with power calculator (UCLA Department of Statistics, available online: http://www.stat.ubc.ca/~rollin/stats/ssize). All data are presented as mean ± SEM. A multiple-sample comparison was applied to test the differences between groups (two-way analysis of variance (ANOVA) with a post-hoc Bonferroni’s multiple comparison test). Statistical significance was considered at *p* < 0.05.

## Figures and Tables

**Figure 1 ijms-20-04621-f001:**
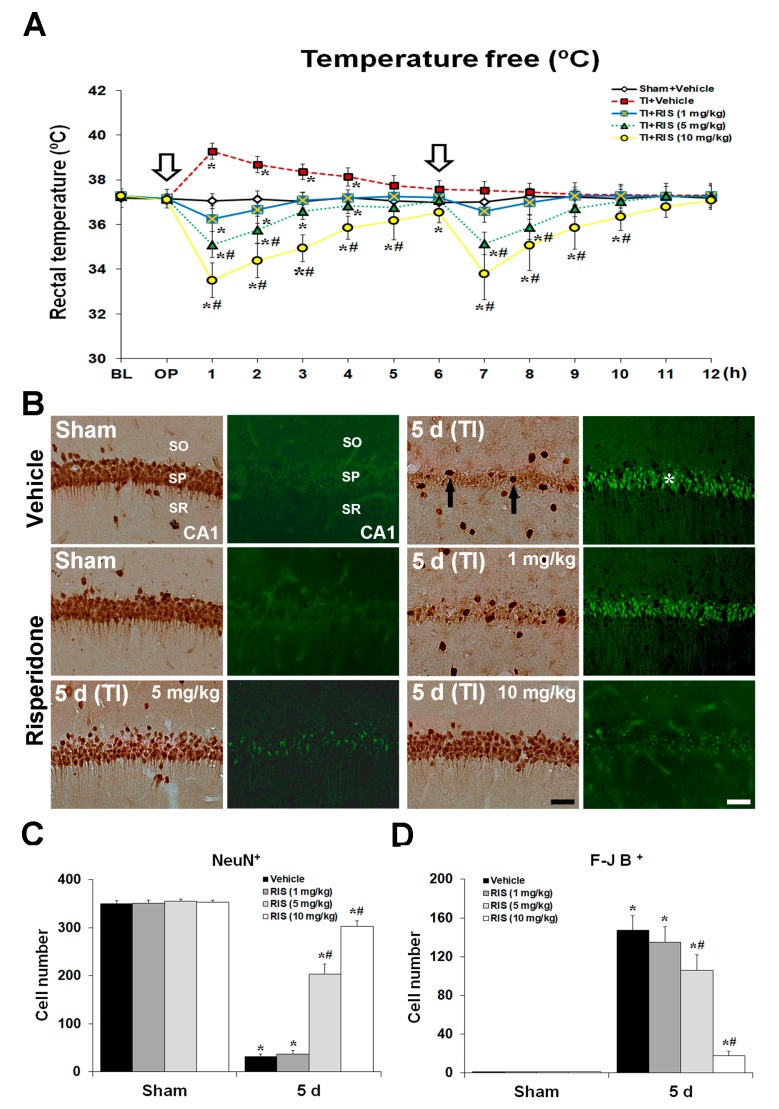
Effects of risperidone (RIS) against transient ischemia (TI) injury under uncontrolled body temperature (UBT) condition. (**A**) Changes in body temperature under UBT condition for 12 h after TI. Body temperature is significantly low in the TI + 10 mg/kg RIS group compared to the TI + vehicle group. White arrows indicate times of RIS treatment. The bars indicate the means ± SEM, *n* = 7/group, two-way analysis of variance (ANOVA) with a post-hoc Bonferroni’s multiple comparison test (* *p* < 0.05 vs. sham + vehicle group; # *p* < 0.05 vs. TI + vehicle group). (**B**) Effects of RIS on NeuN^+^ and F-J B^+^ cells in the CA1 under UBT condition after TI. In the sham + vehicle group, CA1 pyramidal neurons are well stained with NeuN; however, no F-J B^+^ CA1 pyramidal cells are found. In the TI + vehicle group, a few NeuN^+^ cells (arrows) are shown in the stratum pyramidale (SP) 5 days after TI; however, the distribution of NeuN^+^ cells in the TI + RIS group is similar to that in the sham + vehicle group. In the TI + vehicle group, many F-J B^+^ cells (asterisks) are detected in the SP 5 days after TI, and many F-J B^+^ CA1 pyramidal cells (asterisks) are detected; however, in the TI + RIS group, RIS produces a dose-dependent increase in the number of NeuN^+^ CA1 pyramidal neurons, and a dose-dependent decrease in the number of F-J B^+^ CA1 pyramidal cells 5 days after TI. CA1, cornu ammonis 1; CA3, cornu ammonis 3; DG, dentate gyrus; SO, stratum oriens; SR, stratum radiatum. Scale bar = 50 μm. Note histograms of quantitative analyses of NeuN^+^ and F-J B^+^ cells in all the groups, as shown (**C**) and (**D**). The bars indicate the means ± SEM, *n* = 7/group, two-way analysis of variance (ANOVA) with a post-hoc Bonferroni’s multiple comparison test (* *p* < 0.05 vs. sham + vehicle group; # *p* < 0.05 vs. TI + vehicle group).

**Figure 2 ijms-20-04621-f002:**
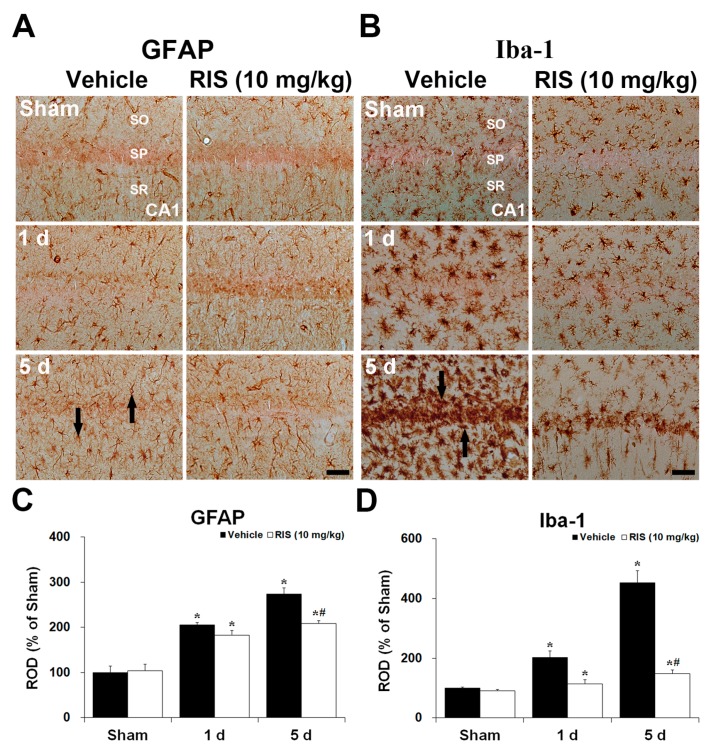
Effects of RIS on GFAP^+^ astrocytes (**A**) and Iba-1^+^ microglia (**B**) in the CA1 after TI under UBT condition. GFAP^+^ astrocytes (arrows in (**A**)) and Iba-1^+^ microglia (arrows in (**B**)) are markedly increased in the TI + vehicle group 5 days after TI; however, their immunoreactivity in the TI + RIS group is significantly lower than that in the TI + vehicle group. SO, stratum oriens; SP, stratum pyramidale; SR, stratum radiatum. Scale bar  =  50 μm. Quantitative analyses of GFAP^+^ (**C**) and Iba-1^+^ cells (**D**). The bars indicate the means ± SEM, *n* = 7/group, two-way analysis of variance (ANOVA) with a post-hoc Bonferroni’s multiple comparison test (* *p* < 0.05 vs. sham + vehicle group; # *p* < 0.05 vs. TI + vehicle group).

**Figure 3 ijms-20-04621-f003:**
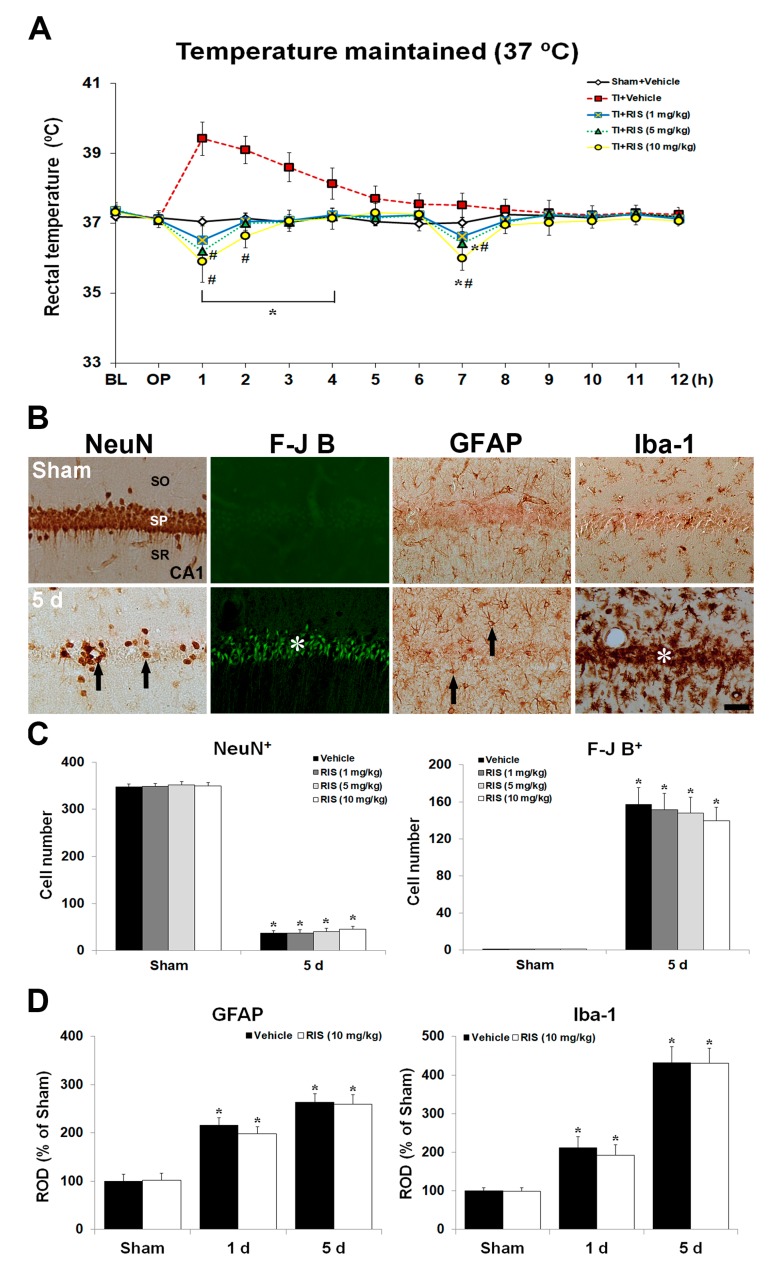
Effects of RIS against TI injury under controlled body temperature (CBT) condition. (**A**) Changes in body temperature under CBT condition for 12 h after TI. Body temperature is slightly low in the TI + 10 mg/kg RIS group compared to the TI + 5 mg/kg RIS group. White arrows indicate times of RIS treatment. The bars indicate the means ± SEM, *n* = 7/group, two-way analysis of variance (ANOVA) with a post-hoc Bonferroni’s multiple comparison test (* *p* < 0.05 vs. sham + vehicle group; # *p* < 0.05 vs. TI + vehicle group). (**B**) Effect of RIS (10 mg/kg) on NeuN^+^, F-J B^+^, GFAP^+^, and Iba-1^+^ cells in the CA1 under CBT after TI 5 days after TI. In the TI + RIS group, a few NeuN^+^ neurons (arrows) and many F-J B^+^ cells (asterisks) are shown in the stratum pyramidale (SP) at 5 days after TI. GFAP^+^ astrocytes (arrows) and Iba-1^+^ microglia (asterisks) are markedly increased in the TI + RIS group under CBT 5 days after TI. SO, stratum oriens; SP, stratum pyramidale; SR, stratum radiatum. Scale bar  =  50 μm. Quantitative analyses of NeuN^+^ and F-J B^+^ cells (**C**), and GFAP^+^ and Iba-1^+^ cells (**D**). The bars indicate the means ± SEM, *n* = 7/group, two-way analysis of variance (ANOVA) with a post-hoc Bonferroni’s multiple comparison test (* *p* < 0.05 vs. sham + vehicle group).

**Figure 4 ijms-20-04621-f004:**
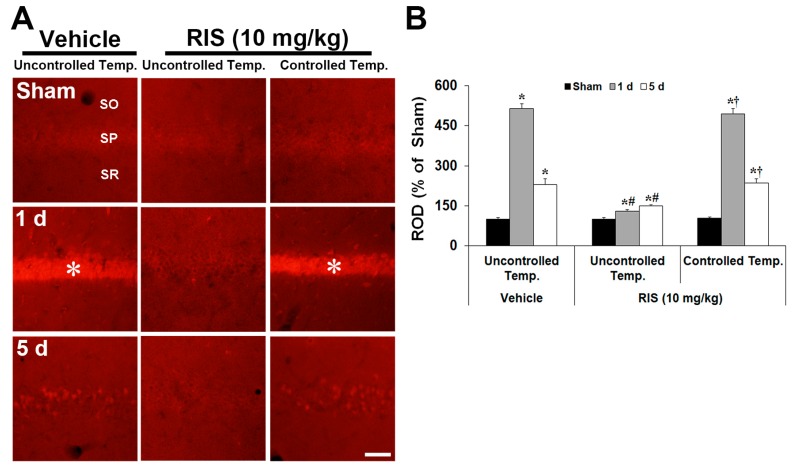
Effects of RIS on superoxide anion production after TI under UBT and CBT conditions. (**A**) Changes of DHE immunofluorescence in the CA1 after TI under UBT and CBT conditions. DHE immunofluorescence is significantly increased in CA1 pyramidal neurons of the stratum pyramidale (SP, asterisks) in the TI + vehicle group under UBT and TI + RIS group under CBT at 1 day after TI; however, DHE immunofluorescence is not significantly increased in the SP of the TI + RIS group under UBT condition. SO, stratum oriens; SR, stratum radiatum. Scale bar = 50 μm. (**B**) Note analyses of DHE immunofluorescence in the SP. A ratio of the ROD is calibrated as %, with the sham + vehicle group designated as 100%. The bars indicate the means ± SEM, *n* = 7/group, two-way analysis of variance (ANOVA) with a post-hoc Bonferroni’s multiple comparison test (* *p* < 0.05 vs. sham + TI + vehicle group under UBT condition; # *p* < 0.05 vs. TI + vehicle group under UBT condition; † *p* < 0.05 vs. TI + RIS group under UBT condition.).

**Figure 5 ijms-20-04621-f005:**
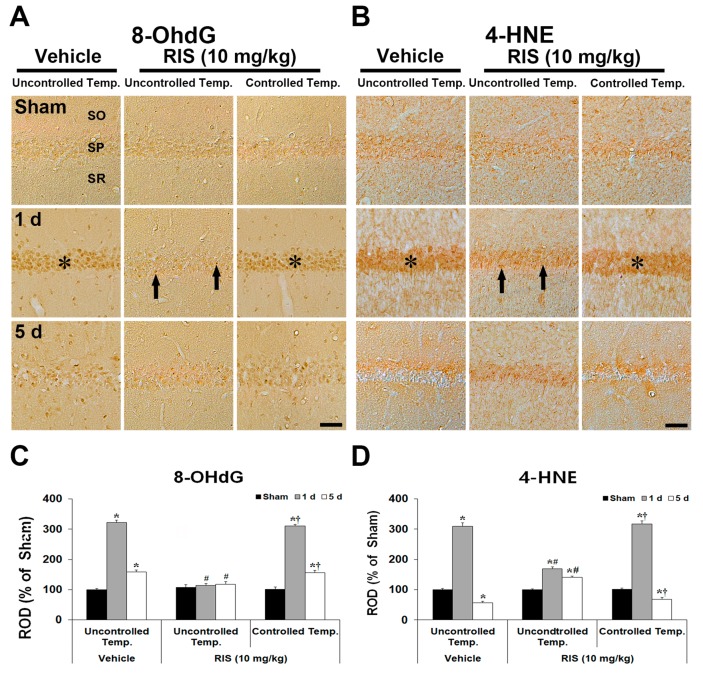
Effects of RIS on 8-OhdG and 4-HNE immunoreactivities after TI under UBT and CBT conditions. Representative images of immunohistochemistry for 8-OhdG (**A**) and 4-HNE (**B**) in the hippocampal CA1 after TI under UBT and CBT conditions. Under UBT condition, 8-OhdG and 4-HNE immunoreactivities in the TI + vehicle group are significantly increased in CA1 pyramidal neurons of the stratum pyramidale (SP, asterisks) in the TI + vehicle group under UBT and TI + RIS group under CBT at 1 day after TI; however, the immunoreactivities are significantly low (arrows) in the TI + RIS group under UBT. Scale bar = 50 μm. Note analyses of 8-OhdG (**C**) and 4-HNE (**D**) immunoreactivity in the SP. A ratio of the ROD is calibrated as %, with the sham + vehicle group designated as 100%. The bars indicate the means ± SEM, *n* = 7/group, two-way analysis of variance (ANOVA) with a post-hoc Bonferroni’s multiple comparison test (* *p* < 0.05 vs. sham + vehicle group under UBT condition; # *p* < 0.05 vs. TI + vehicle group under UBT condition; † *p* < 0.05 vs. TI + RIS group under UBT condition).

**Figure 6 ijms-20-04621-f006:**
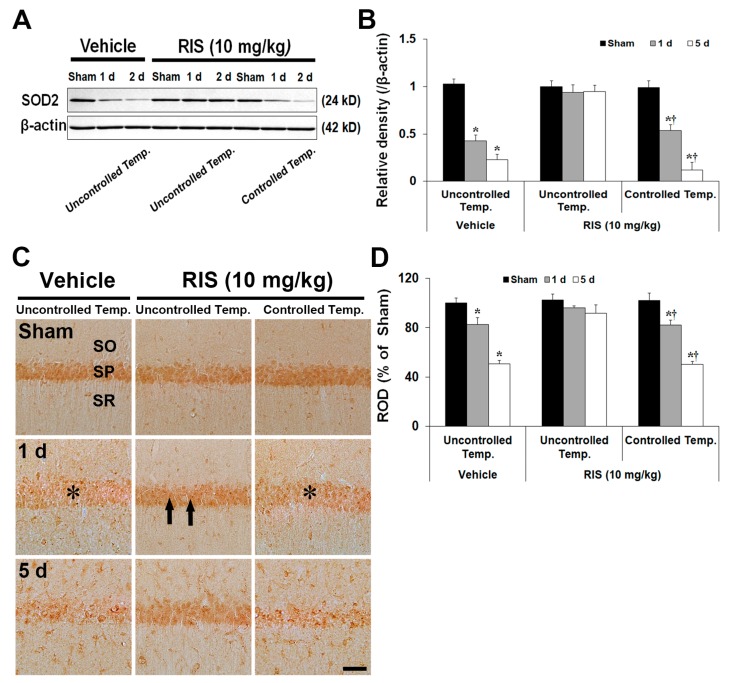
Effects of RIS on levels and immunoreactivity of SOD2 protein after TI under UBT and CBT conditions. (**A**) Western blot analysis of SOD2 protein in the hippocampus after TI under UBT and CBT conditions. Note that SOD2 levels are maintained in the TI + RIS group under UBT after TI. (**B**) Quantitative analysis of SOD2 protein in the hippocampus. Protein expression is normalized to β-actin. The bars indicate the means ± SEM, *n* = 7/group, two-way analysis of variance (ANOVA) with a post-hoc Bonferroni’s multiple comparison test (* *p* < 0.05 vs. sham + vehicle group under UBT condition; † *p* < 0.05 vs. TI + RIS group under UBT condition). (**C**) Representative images of immunohistochemistry for SOD2 in the hippocampal CA1 after TI under UBT and CBT conditions. SOD2 immunoreactivity is significantly decreased in CA1 pyramidal neurons of the stratum pyramidale (SP, asterisks) in the TI + vehicle group under UBT and TI + RIS group under CBT at 1 day after TI; however, the immunoreactivity is maintained (arrows) in the TI + RIS group under UBT. SO, stratum oriens; SR, stratum radiatum. Scale bar = 50 μm. (**D**) Quantitative analysis of SOD2 immunoreactivity in CA1 pyramidal neurons. A ratio of ROD is calibrated as %, with the sham + vehicle group designated as 100%. The bars indicate the means ± SEM, *n* = 7/group, two-way analysis of variance (ANOVA) with a post-hoc Bonferroni’s multiple comparison test (* *p* < 0.05 vs. sham + vehicle group under UBT condition; † *p* < 0.05 vs. TI + RIS group under UBT condition).

**Figure 7 ijms-20-04621-f007:**
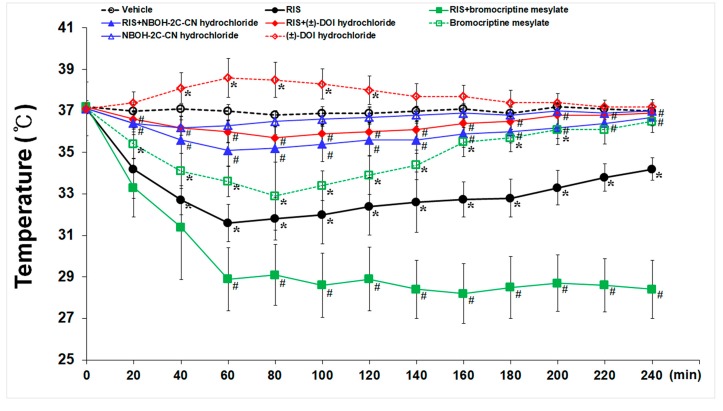
Effects of endogenous 5-HT_2A_- and, D_2_-receptors on RIS-induced hypothermia. Hypothermia was defined as a reduction of at least 2 °C below baseline (normothermia). NBOH-2C-CN hydrochloride does not alter body temperature, but bromocriptine mesylate significant decreased body temperature at 80 min after the administration, and (±)-DOI hydrochloride slightly increases body temperature at 60 min after the administration. Namely, RIS-induced hypothermia is interrupted following the administration of NBOH-2C-CN hydrochloride and (±)-DOI hydrochloride. The bars indicate the means ± SEM, *n* = 7/group, two-way analysis of variance (ANOVA) with a post-hoc Bonferroni’s multiple comparison test (* *p* < 0.05 vs. sham group; # *p* < 0.05 vs. RIS group).

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
