# Peer review of "Risperidone Treatment after Transient Ischemia Induces Hypothermia and Provides Neuroprotection in the Gerbil Hippocampus by Decreasing Oxidative Stress"

_ijms, 2019, doi:10.3390/ijms20184621_

Round 1
Reviewer 1 Report
The present manuscript investigated the neuroprotective effects of risperidone (RIS, an antipsychotic drug) against transient ischemia injury (TI) in gerbils. TI has been shown to induce neuronal death by increasing the production of reactive oxygen species (ROS), which leads to DNA damage and eventually cell death. This study showed that RIS-induced hypothermia is beneficial in IT, and attenuated neuronal loss by decreasing ROS production in a mechanism dependent of 5-serotonin (HT) 2A receptor activation.
In general, this manuscript is well-written and the result section is clear. Mechanistically, the authors show that when gerbils were treated with NBOH-2C-CN, a serotonin receptor agonist, the protective effects of RIS were abolished. Their findings suggested that hypothermia induced by RIS is mediated via 5-HT2A receptor signaling. This is a good manuscript.
Comments:
What population of neurons is responsible for the protective effects of RIS? Are endothelial cells responsible for the protective effects of RIS? Where is 5-HT2AR express in this model? Did the authors observed improvement in cognition after ischemia in gerbils treated with RIS?Author Response
Title: Risperidone treatment after transient ischemia induces hypothermia and provides neuroprotection in the gerbil hippocampus by decreasing oxidative stress
I deeply appreciate the reviewer’s prudent comments on our manuscript.
The responses to the reviewers’ comments are summarized below:
Reviewer #1:
Comment 1: What population of neurons is responsible for the protective effects of RIS?
Response: In cerebral ischemic insults, many studies have described the patter of vulnerability in specific neuronal populations (1). For example, a brief period of complete (global) cerebral ischemia, which can be developed in cardiac arrest, leads to selective neuronal damage/death with changes of glial cells and blood vessels (2). In particular, pyramidal neurons in the hippocampal CA1 region do not die immediately after 5 min of transient cerebral ischemia (TI) in rodents but rather survive over several days after TI. This process is unique and termed “delayed neuronal death (DND)”. The DND of the pyramidal neurons in the hippocampal CA1 region occurs from 4 days after 5 min of TI in rodents (3-5). We have published many papers regarding the DND using F-J B histofluorescence to elucidate the degree of neuronal damage in the hippocampal CA1 region of the gerbil brain after 5 min of TI. F-J B histofluorescence shows a high affinity for degenerating neurons, which has been well established in many studies (6, 7). In this study, we demonstrated the effect of RIS against pathogenesis of CA1 pyramidal cells in the hippocampus following TI.
Heiss WD and Rosner G (1983) Functional recovery of cortical neurons as related to degree and duration of ischemia. Ann Neurol. 14(3):294-301. Pulsinelli WA et al. (1982) Temporal profile of neuronal damage in a model of transient forebrain ischemia. Ann Neurol. 11(5):491-8. Kirino T (1982) Delayed neuronal death in the gerbil hippocampus following ischemia. Brain Res 239:57-69. Crain BJ et al. (1988) Selective neuronal death after transient forebrain ischemia in the Mongolian gerbil: a silver impregnation study. Neuroscience. 27:387-402. Yan BC et al. (2011) Increases of antioxidants are related to more delayed neuronal death in the hippocampal CA1 region of the young gerbil induced by transient cerebral ischemia. Brain Res 24:1425:142-154. Schmued LC and Hopkins KJ (2000) Fluoro-Jade B: a high affinity fluorescent marker for the localization of neuronal degeneration. Brain Res. 874:123-30. Kundrotiene J et al (2004) Fluoro-Jade and TUNEL staining as useful tools to identify ischemic brain damage following moderate extradural compression of sensorimotor cortex. Acta Neurobiol Exp (Wars). 64(2):153-62.
Comment 2: Are endothelial cells responsible for the protective effects of RIS?
Response: This comment is so prudent to evaluate the quality of our manuscript. The novelty and strength of our present study is the morphological evaluation of degenerating/dead neurons in the hippocampal CA1 region using FJB fluorescence staining. For qPCR and western blotting, we used whole hippocampal tissue that contains neurons, glia, vessel, etc. The blood–brain barrier (BBB) is formed by brain endothelial cells that line the cerebral microvasculature, the capillary basement membranes and the endfeet of astrocytes. Reperfusion following cerebral ischemia initiates a cascade of events including inflammation, protease activation, and oxidative and nitrosative stress, all of which increase BBB permeability (1). The increased BBB permeability aggravates hemorrhagic transformation and vasogenic edema, and uncontrolled cerebral edema represents the leading cause of patient mortality within the first week following an ischemic stroke (2). 5-hydroxytryptamine (5-HT) induces potent vasoconstrictor responses in large cerebral vessels mainly through 5-HT receptor activation (3), and it is involved in the control of cerebrovascular tone (4). A previous study has shown that a 5‐HT1A receptor agonist is capable of reducing the increased permeability induced by oxygen-glucose deprivation, and this is inhibited by a 5‐HT1A receptor antagonist (5). However, it is not clear whether 5‐HT2A receptor can reduce endothelial associated brain damage after cerebral ischemia (TI). According to the reviewer’s comment, we are willing to investigate 5-HT2A receptor-mediated BBB dysfunction in rodent brains after a brief period of TI in the future.
Lo EH, Dalkara T, Moskowitz MA (2003). Mechanisms, challenges and opportunities in stroke. Nat Rev Neurosci 4: 399–415. Hacke W, Schwab S, Horn M, Spranger M, De Georgia M, von Kummer R (1996). ‘Malignant’ middle cerebral artery territory infarction: clinical course and prognostic signs. Arch Neurol 53: 309–315. Roon KI, Maassen Van Den Brink A, Ferrari MD, Saxena PR. (1999) Bovine isolated middle cerebral artery contractions to antimigraine drugs. Naunyn Schmiedebergs Arch Pharmacol 360:591–596. Lincoln J. Innervation of cerebral arteries by nerves containing 5-hydroxytryptamine and noradrenaline. Pharmacol Ther. 1995;68(3):473-501 Hind WH, England TJ, O'Sullivan SE. Cannabidiol protects an in vitro model of the blood-brain barrier from oxygen-glucose deprivation via PPARγ and 5-HT1A receptors. Br J Pharmacol. 2016 Mar;173(5):815-25.
Comment 3: Where is 5-HT2AR express in this model?
Response: Serotonergic neurons was first discovered in the brainstem by Dahlström and Fuxe in 1964 (1). They release 5-HT throughout the CNS (2, 3) as expected after the brain serotonin discovery (4). 5-HT cell bodies are mainly localized in the raphe nuclei with their axons innervating almost brain regions (5). The hippocampus is a principal target of serotonergic afferents along with the limbic system (6). All serotonin receptor families are remarkably expressed in the hippocampus, which is a part of the limbic system, and its structures are related with memory processing, emotional association with memory, judgment, affect, and motivation or the organization of planned actions (6). The innervation of serotonergic pathways in the hippocampus and the diverse expression of serotonin receptors in this brain area reflects overall functions related to 5-HT, in particular with cognition, mood and food intake. In addition, Julius et al. (1990) found an encoding sequence for 5-HT2 which was expressed in the hippocampus in a 10-fold lower level than in the rat cortex (7). 5-HT2A expression in the human hippocampus was confirmed with RT-PCR technique (8). Immunoreactivity for 5-HT2A receptor in the hippocampus was found primarily in the pyramidal cell layer of CA1–CA3 and in the granular layer of the dentate gyrus (9). The precise subcellular distribution of 5-HT2A receptor within the hippocampal subfields has not been defined yet. According to the reviewer’s comment, we are willing to investigate the expression of 5-HT2A receptor in the gerbil hippocampus after 5 min of TI in the future.
A. Dahlström and K. Fuxe, “Localization of monoamines in the lower brain stem,” Experientia, vol. 20, no. 7, pp. 398–399, 1964. H. W. Steinbusch, “Distribution of serotonin-immunoreactivity in the central nervous system of the rat-cell bodies and terminals,” Neuroscience, vol. 6, no. 4, pp. 557–618, 1981. W. Wisden, “Cre-ating ways to serotonin,” Frontiers in Neuroscience, vol. 4, p. 167, 2010. B. M. Twarog and I. H. Page, “Serotonin content of some mammalian tissues and urine and a method for its determination,” The American Journal of Physiology, vol. 175, no. 1, pp. 157–161, 1953. Y. Charnay and L. Léger, “Brain serotonergic circuitries,” Dialogues in Clinical Neuroscience, vol. 12, no. 4, pp. 471–487, 2010.
6 J. G. Hensler, “Serotonergic modulation of the limbic system,” Neuroscience and Biobehavioral Reviews, vol. 30, no. 2, pp. 203–214, 2006.
D. Julius, K. N. Huang, T. J. Livelli, R. Axel, and T. M. Jessell, “The 5HT2 receptor defines a family of structurally distinct but functionally conserved serotonin receptors,” Proceedings of the National Academy of Sciences of the United States of America, vol. 87, no. 3, pp. 928–932, 1990. P. W. Burnet, S. L. Eastwood, and P. J. Harrison, “Detection and quantitation of 5-HT1A and 5HT(2A) receptor mRNAs in human hippocampus using a revel-se transcriptase-polymerase chain reaction (RT-PCR) technique and their correlation with binding site densities and age,” Neuroscience Letters, vol. 178, no. 1, pp. 85–89, 1994. Q. H. Li, K. Nakadate, S. Tanaka-Nakadate, D. Nakatsuka, Y. Cui, and Y. Watanabe, “Unique expression patterns of 5-HT2A and 5-HT2C receptors in the rat brain during postnatal development: western blot and immunohistochemical analyses,” Journal of Comparative Neurology, vol. 469, no. 1, pp. 128–140, 2004.
Comment 4: Did the authors observed improvement in cognition after ischemia in gerbils treated with RIS?
Response: We previously reported that a drug administration after TI improved ischemia-induced short-term and spatial learning memory impairment using the passive avoidance and Barnes maze tests in gerbils subjected to TI (1–3). We are convinced the improvement in cognition after ischemia in gerbils treated with RIS. In addition, it is important to study how RIS recovers cognitive impairment after TI. In this regard, we are studying the mechanism of improvement of impaired cognition following TI in gerbils after treatment with RIS.
Chen BH, Park JH, Lee YL, Kang IJ, Kim DW, Hwang IK, Lee CH, Yan BC, Kim YM, Lee TK, Lee JC, Won MH, Ahn JH. Melatonin improves vascular cognitive impairment induced by ischemic stroke by remyelination via activation of ERK1/2 signaling and restoration of glutamatergic synapses in the gerbil hippocampus. Biomed Pharmacother. 2018 Dec;108:687-697. Chen BH, Ahn JH, Park JH, Song M, Kim H, Lee TK, Lee JC, Kim YM, Hwang IK, Kim DW, Lee CH, Yan BC, Kang IJ, Won MH. Rufinamide, an antiepileptic drug, improves cognition and increases neurogenesis in the aged gerbil hippocampal dentate gyrus via increasing expressions of IGF-1, IGF-1R and p-CREB. Chem Biol Interact. 2018 Apr 25;286:71-77 Ahn JH, Chen BH, Shin BN, Cho JH, Kim IH, Park JH, Lee JC, Tae HJ, Lee YL, Lee J, Byun K, Jeong GB, Lee B, Kim SU, Kim YM, Won MH, Choi SY. Intravenously Infused F3.Olig2 Improves Memory Deficits via Restoring Myelination in the Aged Hippocampus Following Experimental Ischemic Stroke. Cell Transplant. 2016 Dec 13;25(12):2129-2144.
We deeply thank again for the prudent comments on our manuscript.
Very sincerely yours

Reviewer 2 Report
In this manuscript, authors demonstrated the function and mechanism of Risperidone in maintaining the hypothermia and protection of hippocampal neurons in transient ischemia condition. The findings are very compelling and generate good impact in the field.
These are the minor comments to be addressed.
Title: Title should be short enough to get the essence of findings
Risperidone treatment after transient ischemia induces hypothermia and provides neuroprotection in the gerbil hippocampus
Figure 1B: Very confusing- must be labeled well- each panel must be denoted with different letter- explain well in legend.
Statistical significance for bar graphs must be presented in better way. It is confusing to compare with different groups.
Results: combined both section 2.2.1 and 2.2.2
Rational behind checking the levels of SOD2 must be explained well.
Author Response
Title: Risperidone treatment after transient ischemia induces hypothermia via and provides neuroprotection in the gerbil hippocampus by decreasing oxidative stress
I deeply appreciate the reviewer’s prudent comments on our manuscript. The responses to the reviewers’ comments are summarized below:
Reviewer #2:
Comment 1: Title: Title should be short enough to get the essence of findings.
Response: According to the reviewer’s comment, we deleted “via 5-HT2a antagonism” in the title.
Comment 2: Figure 1B: Very confusing- must be labeled well- each panel must be denoted with different letter- explain well in legend.
Response: This comment is so prudent to elevate the quality of our MS. According to the reviewer’s comment, we newly edited the text for Figure 1B, labelling for Figure 1B, and its legend as follows: “(B) Effects of RIS on NeuN+ and F-J B+ cells in the CA1 under UBT condition after transient cerebral ischemia (TI). In the sham+vehicle group, CA1 pyramidal neurons are well stained with NeuN; however, no F-J B+ CA1 pyramidal cells are found. In the TI+vehicle group, a few NeuN+ cells (arrows) are shown in the stratum pyramidale (SP) 5 days after TI, and many F-J B+ CA1 pyramidal cells (asterisk) are detected; however, in the TI+RIS group, RIS produces a dose-dependent increase in the number of NeuN+ CA1 pyramidal neurons, and a dose-dependent decrease in the number of F-J B+ CA1 pyramidal cells 5 days after TI. CA1, cornu ammonis 1; CA3, cornu ammonis 3; DG, dentate gyrus, SO, stratum oriens; SR, stratum radiatum. Scale bar = 50 μm. Note histograms of quantitative analyses of NeuN+ and F-J B+ cells in all the groups, as shown C and D. The bars indicate the means ± SEM (*p < 0.05 vs. sham+vehicle group; #p < 0.05 vs TI+vehicle group).”
Comment 3: Statistical significance for bar graphs must be presented in better way. It is confusing to compare with different groups.
Response: I am so sorry that we made the reviewer confused. According to the reviewer’s comment, we fixed them. Please refer to the figures.
Comment 4: Results: combined both section 2.2.1 and 2.2.2
Response: We fully agree with the reviewer’s comment. However, it is not possible to combine both section 2.2.1 and 2.2.2, because these two contents have different results. Please comprehend our opinion.
Comment 5: Rational behind checking the levels of SOD2 must be explained well.
Response: This comment is so prudent to elevate the quality of our MS. As the reviewer pointed out, we newly inserted the expression rates of the levels of SOD2.
We deeply thank again for the prudent comments on our manuscript.
Very sincerely yours,
